# Myopia Is Suppressed by Digested Lactoferrin or Holo-Lactoferrin Administration

**DOI:** 10.3390/ijms24065815

**Published:** 2023-03-18

**Authors:** Yifan Liang, Shin-ichi Ikeda, Junhan Chen, Yan Zhang, Kazuno Negishi, Kazuo Tsubota, Toshihide Kurihara

**Affiliations:** 1Laboratory of Photobiology, Keio University School of Medicine, 35 Shinanomachi, Shinjuku-ku, Tokyo 160-8582, Japan; 2Department of Ophthalmology, Keio University School of Medicine, 35 Shinanomachi, Shinjuku-ku, Tokyo 160-8582, Japan; 3Tsubota Laboratory, Inc., 34 Shinanomachi, Shinjuku-ku, Tokyo 160-0016, Japan

**Keywords:** myopia, sclera, choroid, lactoferrin, lactoferricin, holo-lactoferrin

## Abstract

Myopia is becoming a leading cause of vision impairment. An effective intervention is needed. Lactoferrin (LF) is a protein that has been reported to inhibit myopia progression when taken orally. This study looked at the effects of different forms of LF, such as native LF and digested LF, on myopia in mice. Mice were given different forms of LF from 3 weeks of age, and myopia was induced with minus lenses from 4 weeks of age. Results showed that mice given digested LF or holo-LF had a less elongated axial length and thinned choroid, compared to those given native-LF. Gene expression analysis also showed that the groups given native-LF and its derivatives had lower levels of certain cytokines and growth factors associated with myopia. These results suggest that myopia can be more effectively suppressed by digested LF or holo-LF than native-LF.

## 1. Introduction

Myopia has become significantly more prevalent over the past few decades [1,2,3]. As an estimate of the global prevalence of myopia, the prevalence of myopia and high myopia is expected to rise significantly on a global scale, affecting almost 5 billion and 1 billion people, respectively, by the year 2050 [2]. Myopia is becoming the leading cause of vision impairment worldwide [2,3]. The predominant cause of myopia is axial elongation, which means that light from distant objects is focused in front of the retina [1]. The severity of myopia can be categorized by the appearance of two parameters. A spherical equivalent refractive error of −0.50 diopters (D) or less is the current consensus threshold value for myopia, while a spherical equivalent refractive error of −6.00 D or less become the threshold value for high myopia [4], which can increase the risk vision impairment. Low myopia, with a refractive error between –6 D and −0.5 D, has an axial length between 24 mm and 26 mm, while high myopia, with a refractive error lower than –6 D, has an axial length longer than 26 mm [4,5,6]. Reports have consistently shown a negative relationship between axial length and myopia, indicating that longer axial lengths are associated with more severe myopia [7,8,9]. Owing to morphological changes, physical stimuli such as stress are applied to the posterior segment of the eye, resulting in visual impairment, such as myopic macular degeneration (MMD), retinal detachment (RD), cataract, and open angle glaucoma (OAG), and macular optic neuropathy [3,10,11]. The visual system comprises the extraocular muscles, which play a vital role in both moving the eyes and stabilizing the resulting images [12,13]. Myopia has also been proven to be associated with masticatory muscle bioelectrical activity and muscle thickness. In addition, myopia in Singapore, for example, costs SGD 959 (USD 755) million annually, resulting in significant out-of-pocket expenses that burden patients and their families; thus, myopia has a notable public health and economic impact [14]. Consequently, intervention in axial length is required. Inflammation and inflammation-associated degradation of extracellular matrix (ECM) have been reported to be involved in axial elongation and myopia [15,16,17]. Thus, anti-inflammatory substances may suppress the progression of myopia by reducing scleral remodeling.

Lactoferrin (LF) is an iron-binding glycoprotein distributed in secretions such as breast milk, tear fluid, saliva, blood, mucus, and neutrophils, and is necessary for protection against infection and inflammation protection [18,19,20,21]. In addition to unmodified lactoferrin (native-LF, nLF), other derivatives of LF have anti-inflammatory effects, including digested LF (LF hydrolysates) and iron-saturated LF (Holo-LF). Lactoferrin hydrolysates contain many peptides of varying lengths depending on the digestive enzymes [22]. Lactoferricin (LFcin) is a well-known LF hydrolysate that is hydrolyzed by pepsin [23,24]. Holo-lactoferrin is also known to have different physiological effects, such as different effects on the iron chelate effect owing to its high iron saturation compared with LF [25]. A previous study showed that oral LF administration inhibited minus lens-induced myopia (LIM) development in mice by suppressing inflammation, such as increased interleukin (IL)-6 and matrix metalloprotease (MMP)-2 activity [26]. This previous study showed that LF can suppress myopia progression; however, whether the inhibitory effects on myopia were owing to the different iron saturation levels or LF derivatives was unclear. To investigate the differences in the suppressive effects of native-LF and LF derivatives on myopia progression, we designed the present study using the described LF products. We predicted that the LF derivatives could suppress the myopia progression more efficiently.

## 2. Results

### 2.1. Changes in Refraction, AL, and CT of Digested-LF-Administered LIM Mice

The control group, which was only administered solvent, showed significant negative refraction shifts (*p* < 0.001), elongated axial length (*p* < 0.01), and thinned choroid (*p* < 0.05) (Figure 1A–C) in their −30 D LIM eyes, which indicates significant myopia shifts. Significant differences were also found in refraction (*p* < 0.001) between the 0 D and −30 D eyes among all treated groups (Figure 1A–C). Compared with the control group, the −30 D LIM eyes of all treated groups showed significantly less negative refraction shift (*p* < 0.001), short AL, and thick choroid (Figure 1A–C), suggesting that myopia shifts were suppressed in the treated groups. Figure 1D–F show the left-right delta of refraction, AL, and CT changes. Compared to the control group, all treated groups showed significantly reduced refractive changes (*p* < 0.001). In addition, the pepsin + LF group showed significantly shorter AL changes (*p* < 0.05) (Figure 1E) and significantly fewer CT changes (*p* < 0.01) (Figure 1F) than the native-LF and trypsin + LF groups did. These results suggest that administration of native-LF and artificially digested LF can suppress myopia progression, and administration of pepsin + LF results in a particularly pronounced suppression of myopia shift.

### 2.2. Changes in the Inflammatory Status of the Choroid and Sclera owing to Oral Administration of LF and Its Derivatives

Myopia progression is associated with inflammatory response [16,17]. Interleukin-1 and TNF-α can activate NF-kB, an important transcription factor that regulates inflammatory response [27]. Matrix metalloprotease-2, a target of inflammatory signaling via NF-κB, causes scleral remodeling [28], which in turn leads to axial elongation. Therefore, we assessed the expression levels of IL-1, tumor necrosis factor (TNF)-α, and MMP-2 in the sclera. Figure 2A shows that in the PBS control group, −30 D LIM eyes had significantly higher TNF-α (*p* < 0.001) and MMP-2 (*p* < 0.001) mRNA expression levels than the 0 D eyes did, whereas these levels were lower in the groups administered native-LF and its hydrolysates than in the PBS group. This suggests that native-LF and LF derivatives can reduce the inflammatory response to myopia progression in the sclera.

Upregulation of IL-6 activates NF-κB and induces monocyte macrophage infiltration, which leads to inflammation [17]. Interleukin-8, a downstream target of NF-kB, is a chemokine and an IL produced by macrophages and vascular endothelial cells [29]. These ILs can serve as indicators of the inflammatory response. High myopia is also associated with severe fundus thinning. It is hypothesized that a thin choroid creates hypoxic conditions in the basal lamina. Vascular endothelial growth factor is a dimeric glycoprotein that stimulates neovascularization [30]. Studies have shown that low levels of vascular endothelial growth factor (VEGF) A expression are associated with choroidal thinning, decreased choroidal vessels, and AL elongation [31,32]. Figure 2B shows that in the PBS control group, the choroid of −30 D LIM eyes had significantly higher IL-8 (*p* < 0.01) mRNA expression than that of 0 D eyes, whereas the expression levels were low in the group administered native-LF and its hydrolysates. These results suggest that treatment with native-LF and its derivatives can reduce the inflammatory response during myopia progression in the choroid.

### 2.3. Changes in Refraction, AL, and CT of Holo-LF-Administered LIM Mice

The −30 D LIM eyes of the control group, which received only solvent administration, exhibited significantly negative refraction shifts (*p* < 0.001), elongated AL (*p* < 0.05), and thinning choroid (*p* < 0.05) (Figure 3A–C), indicating significant myopia changes. There were also significant variations in refraction (*p* < 0.001) between the 0 D and −30 D eyes in all treatment groups. Compared to the control group, the −30 D LIM eyes of all treated groups showed no significant change in AL and choroid thickness (Figure 3A–C), indicating that myopia shifts were controlled in the treated groups.

Figure 3D,E illustrate the left-right delta of refraction, AL, and CT changes. All treated groups showed significantly reduced refraction changes (*p* < 0.001) compared to the control group. In addition, the Holo-LF group showed significantly fewer CT alterations (*p* < 0.05) than the native-LF group (Figure 3F). These results revealed that the administration of native-LF and Holo-LF can inhibit the progression of myopia, with Holo-LF having a greater effect on inhibiting choroidal thinning.

### 2.4. Changes in the Inflammatory Status of the Choroid and Sclera following Oral Administration of LF and Holo-LF

In the PBS control group, −30 D LIM eyes had significantly higher TNF-α (*p* < 0.01) mRNA expression than the 0 D eyes did, whereas no significant differences were observed in the native- and Holo-LF-administered groups (Figure 4A), suggesting that native-LF and Holo-LF can reduce the inflammatory response of myopia progression in the sclera. As shown in Figure 4B, in the PBS control group, −30 D LIM eyes expressed significantly more IL-8 (*p* < 0.001) mRNA than did 0 D eyes, although these expression levels were lower in the native- and Holo-LF-administered groups. Additionally, in the native- and Holo-LF-administered groups, significantly elevated VEGFA (*p* < 0.01) mRNA expression was observed in −30 D LIM eyes compared to that in 0 D eyes, but in the PBS control group, similar changes were not observed. These results show that native-LF and its derivatives can attenuate the inflammatory response to the evolution of myopia in the choroid.

## 3. Discussion

In this study, orally administered LFcin, a derivative of LF digested by pepsin, appeared to provide a more effective treatment than LF did for improving myopia progression in mice. Oral native-LF has been shown to be effective in downregulating MMP-2 expression in the sclera to suppress the progression of myopia [26]. The visual system, including the extraocular muscles, plays a critical role in both moving the eyes and stabilizing resulting images [12,13]. Recent studies suggest that myopia may be associated with masticatory muscle bioelectrical activity and muscle thickness. On the other hand, LF may act as a functional factor that maintains the proliferation of stem cells (SCs) and promotes myoblast proliferation by activating the extracellular signal-regulated kinase (ERK)1/2 signaling pathway, at least partially through LRP1, and inducing the differentiation of myoblasts into myotubes. In addition, LF promotes myotube hypertrophy. LF depletion impaired the proliferation of SCs by downregulating ERK1/2 signaling [33,34]. Moreover, intraperitoneal injection of exogenous recombinant lactoferrin (R-LF) was found to effectively help repair and renew skeletal muscle after injury. Thus, LF may suppress myopia progression by repairing the function of the muscular system [33,34]. We found that in both axial elongation and choroidal thinning, LFcin showed more significant suppressive effects than those observed in the groups treated with other LF products, suggesting that regarding myopia suppression, LFcin can show a stronger effect than native-LF.

Results of the real-time PCR suggested inhibition of inflammation in the choroid and sclera. The immune-regulating properties of LF, such as cell migration, proliferation, and modulation of the inflammatory response [35], have also been reported in wound healing activity [36,37,38], via hyaluronan synthesis [21]. Although the in vivo kinetics of LF have not been fully elucidated, based on previous studies, LFcin exemplifies a stronger immune upregulation than does LF [39]. Lactoferricin, which contains the N-terminal region of LF, is involved in several LF functions [40]. Furthermore, it was reported that peptides from LFcin can stimulate antimicrobial activity of neutrophils via peptides LFcin B 17–26 or inhibit the proliferation of MethA and MT-1 tumor cell lines via peptides LFcin B 13–31 [41], which peptides from LF cannot. These immune-regulating properties may explain the lower expression levels of IL and TNF-α in the pepsin + LF group than those in the LF group.

Based on our results, myopia-induced eyes showed fewer axial and choroidal changes in both the native- and Holo-LF-administered groups than in the PBS-administered group. These results suggest that both native-LF and Holo-LF can suppress myopia progression. Interestingly, in addition to the inhibitory effect of native-LF on myopia development, Holo-LF effectively interferes with choroidal thinning.

Kawashima et al. [42] reported that administration of LF to mice suppressed lacrimal gland inflammation and dry eye symptoms. Subsequently, in a study [43] to explore the mechanism of dry eye treatment, LF was observed to regulate the composition of the intestinal microbiome, and administration of LF markedly increased short-chain fat acid concentrations. They reported that these results led to increased expression of immune regulatory Treg cells and changes in cytokine profiles, which may lead to the control of systemic inflammation and subsequent protection of lacrimal gland function [44]. These studies have shown that LF is effective against dry eye disease. The relationship between dry eye and myopia has already been reported [5,6], and choroidal changes have been reported [45] as being associated with suppression of myopia progression and dry eye progression. Furthermore, an epidemiological study showed that patients with choroidal thinning had low levels of iron, copper, and calcium ions in the blood [46]. Lactoferrin may be effective for myopia because, as these studies suggest, it can be a common denominator in the etiology of dry eye disease and myopia. In addition, Zhang et al. [47] found that when MDA-MB-231 cells, a type of triple-negative breast cancer, were cultured in a hypoxic environment, Holo-LF catalyzed the decomposition of a significant amount of H_2_O_2_ and improved the hypoxic microenvironment. This study showed that Holo-LF is effective in reducing ischemia, which may explain why Holo-LF administration can suppress the change in choroidal thickness during myopia progression. Based on the above reports, the ability of Holo-LF to inhibit choroidal thinning reported in the present study may have potential for use in orally administered myopia progression suppression therapy.

In conclusion, we showed that oral administration of LFcin can suppress myopia more effectively than administration of native-LF. Additionally, Holo-LF can suppress choroidal thinning more effectively than native-LF with low saturation. This study provides positive evidence for the development of supplements to prevent myopia.

## 4. Materials and Methods

### 4.1. Preparation of Native-LF, LF Hydrolysates, and Holo-LF

Lactoferrin powder (LF, from bovine milk 25 g; Fuji Film Wako Chemicals, Osaka, Japan) was used to prepare the native-LF solution (160 mg/mL in phosphate-buffered saline (PBS)). Lactoferrin hydrolysates were prepared using pepsin (1:10,000, stable powder 100 g, Sigma-Aldrich, Burlington, MA, USA) and trypsin (1:10,000, Type XI, lyophilized powder, ≥6000 BAEE units/mg protein 500 MG, Sigma-Aldrich, Burlington, MA, USA), and the solution was incubated for 4 h at 37 °C. The pepsin-treated solution was adjusted to pH 2–2.5, and the trypsin-treated solution was adjusted to pH 7.5–8 before incubation [22]. After incubation, the pepsin- and trypsin-treated solutions were adjusted to pH 6.5–7 and 5.5–6, respectively, to stop the enzyme reaction [25]. All the solutions were prepared at 160 mg/mL.

### 4.2. Animals

All animal experiments in this study were approved by the Animal Experimental Committee of Keio University (permit number: 16017-3). Our study adhered to the Institutional Guidelines on Animal Experimentation at Keio University, the Association for Research in Vision and Ophthalmology (ARVO) Statement for the Use of Animals in Ophthalmic and Vision Research, and the Animal Research: Reporting of In Vivo Experiments (ARRIVE) guidelines for the use of animals in research. Male C57BL/6J mice (3-week-old) were purchased from CLEA Japan (Yokohama, Japan). Five mice were maintained in one cage and allowed free intake of standard chow and water. The mice were housed in a 12 h/12 h light/dark cycle environment (the dark cycle extended from 8:00 p.m. to 8:00 a.m.) at 23 ± 3 °C. In each group, five mice were used in the experiment, and the whole experiment was repeated once to ensure reproducibility. This sample number is calculated based on previous study [26] with a 2.67 effect size and a 95% confidence interval by G Power (Version 3.1.9.7, University Dusseldorf, Düsseldorf, Germany). The light cycle was maintained using a 50-lux background according to previous reports on experimental myopia induction [48,49].

#### 4.2.1. LF Administration

Lactoferrin, pepsin-hydrolyzed LF, trypsin-hydrolyzed LF, and Holo-LF were administered to mice by gastric sonde at 4 p.m. to 7 p.m. once a day from 3 to 7 weeks of age. All reagents were administered at a dose of 1600 mg/kg body weight.

#### 4.2.2. LIM in Mice

Lens-induced myopia was induced in mice at 4 weeks of age, as reported in previous studies (28, 29). Midazolam (Sandoz K.K., Tokyo, Japan), medetomidine (Domitor, Orion Corporation, Espoo, Finland), and butorphanol tartrate (Meiji Seika Pharma Co., Ltd., Tokyo, Japan) were used to induce general anesthesia in mice. The scalp was excised to expose a 0.8 cm^2^ area of the skull, and the periosteum was etched away. The eyeglasses were then affixed to the mouse’s head using a self-curing dental adhesive system (Super-Bond; Sun Medical, Shiga, Japan). The eyeglasses were manufactured using a three-dimensional printer and were designed specifically for mice. The eyeglasses have a hinged component that allows the position of the left and right frames to be adjusted according to the shape of the mouse skull or removed for cleaning. The lenses of the eyeglasses were designed by the Rainbow Optical Research Institute (Tokyo, Japan) based on human hard-contact lenses. As an internal control, all left sides of the eyeglasses used in this study were fitted without a lens, whereas the right sides were fitted with −30 D lenses. The eyeglasses of each mouse were removed at least twice a week for cleaning.

#### 4.2.3. Refraction, Axial Length (AL), and Choroid Thickness (CT) Measurements

Refractions and ALs were measured as previously described (28, 29). Briefly, the refractive conditions were measured using an infrared photorefractor (Steinbeis Transfer Centre, Karlsruhe, Germany). To ensure mydriasis and cycloplegia, a solution of tropicamide and phenylephrine hydrochloride (Mydrin-P ophthalmic solution; Santen Pharmaceutical, Osaka, Japan) was applied to the mouse eye 5 min before the measurement. Refractions were obtained along the optic axis under general anesthesia induced by midazolam. After measuring refraction, the AL and CT were examined using a spectral domain-optical coherence tomography (SD-OCT) system (Envisu R4330; Leica, Hessen, Germany) that was optimized for mice. Axial length was defined as the distance between the corneal vertex and retinal pigment epithelium (RPE) layer next to the optic nerve. In accordance with a previous report, the CT was assessed using SD-OCT equipment (32). ImageJ software version 1.52v11 (National Institutes of Health, Bethesda, MD, USA) was used to quantify the area of the circumference (0.5 mm) from the disk circled at the RPE boundary and the posterior surface of the choroid. The average CT was estimated by dividing the area by the circumference. Measurements of refraction, AL, and CT were performed twice for each mouse before initiation of LIM (0 W) and at 3 weeks (3 W) later. The relative differences between eyes were calculated as follows: the average value of each eye at 3 W minus the average value at 0 W in each group. The delta differences were calculated as the relative differences of the right eyes minus the relative differences of the left eyes.

### 4.3. RNA Expression Test

#### 4.3.1. Sampling

Mice were euthanized using the cervical dislocation method. Subsequently, the eyes were enucleated and immersed in PBS. The eyes were cut along the edge of the cornea/sclera, and the cornea, lens, and retina were removed. The sclera/choroid was cut and placed flat on a glass slide. The sclera and choroid were carefully separated using a blade, and the samples were placed in a pre-prepared 2 mL tube. The collected samples were frozen in liquid nitrogen and stored in a −80 °C deep freezer until use.

#### 4.3.2. RNA Extraction

An RNeasy Micro Kit (Qiagen, Venlo, The Netherlands) was used to extract RNA from the mouse sclera and choroid. All processes were followed the guidelines which was including in the kit.

#### 4.3.3. Reverse Transcription-Polymerase Chain Reaction (RT-PCR)

Extracted mRNA (6 μL) was incubated at 65 °C for 5 min. A cDNA synthesis kit (Toyobo, Osaka, Japan) was used to reverse transcribe the mRNA to cDNA. Genomic DNA remover (2 μL) was added to each sample and incubated in 37 °C for 5 min. A cDNA synthesis solution (2 μL) was added to each sample and incubated following the manufacturer’s instructions to apply the reverse transcription reaction.

QuantStudio 5 (Applied Biosystems, Waltham, MA, USA) was used to perform the RT-PCR in the Fast Program with primers (Thermo Fisher Scientific, Waltham, MA, USA) for MMP-2, IL-6, IL-8, vascular endothelial growth factor-A (VEGFA), and tumor necrosis factor-α (TNF-α).

The primer sequences used were:

β-actin Forward: 5′-GGAGGAAGAGGATGCGGCA-3′

β-actin Reverse: 5′-GTGGCAGCTCTTCTCAAAGC-3′

IL1 Forward: 5′-CAACCAACAAGTGATATTCTCCATG-3′

IL1 Reverse: 5′-GATCCACACTCTCCAGCTGCA-3′

IL-6 Forward: 5′-CTACCCCAATTTCCAATGCT-3′

IL-6 Reverse: 5′-ACCACAGTGAGGAATGTCCA-3′

IL-8 Forward: 5′-AGACTCCAGCCACACTCCAA-3′

IL-8 Reverse: 5′-TGACAGCGCAGCTCATTG-3′

MMP-2 Forward: 5′-CAAGTTCCCCGGGAT-3′

MMP-2 Reverse: 5′-TTCTGGTCAAGGTCAC-3′

VEGFA Forward: 5′-CTCCAGGGCTTCATCGTTA-3′

VEGFA Reverse: 5′-CAGAAGGAGAGCAGAAGTCC-3′

TNF-α Forward: 5′-CTGTAGCCCACGTCGTAGC-3′

TNF-α Reverse: 5′-TTGAGATCCATGCCGTTG-3′ 

### 4.4. Statistical Analyses

Data were output using QuantStudio 5 software (Applied Biosystems, Waltham, MA, USA), input into laboratory computers, and analyzed using SPSS Version.28 (IBM, Armonk, NY, USA) and Microsoft Office 365 Excel (Microsoft Corp, Seattle, WA, USA). Qualitative data are described as a ratio (−30 D/0 D). The two-tailed Student’s *t*-test was used for the left–right eye comparison analysis. Comparisons between groups in each experiment were analyzed using analysis of variance (ANOVA) and Tukey’s post hoc test. Statistical significance was set at *p* < 0.05.

## Figures and Tables

**Figure 1 ijms-24-05815-f001:**
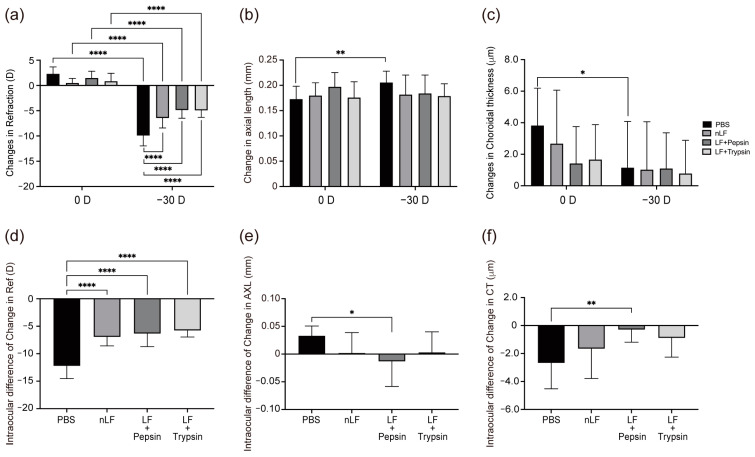
Changes in refraction, axial length (AL), and choroidal thickness (CT) in lens-induced myopia (LIM) mice administered native-lactoferrin (nLF) and its derivatives. (**a**) Changes in refraction in 0 D and −30 D eyes over a 3-week period. (**b**) Changes in AL in 0 D and −30 D eyes over a 3-week period. (**c**) Changes in choroidal thickness in 0 D and −30 D eyes over a 3-week period. (**d**) Delta between the refractive changes in 0 D and −30 D eyes over a 3-week period. (**e**) Delta between the AL changes in 0 D and −30 D eyes over a 3-week period. (**f**) Delta between the choroidal thickness changes in 0 D and −30 D eyes over a 3-week period. n = 10 per group. * *p* < 0.05, ** *p* < 0.01, and **** *p* < 0.0001. The two-tailed Student’s *t*-test was used for left-right comparison analysis. Comparisons between PBS, LF, pepsin + LF, and trypsin + LF groups in each experiment was analyzed by analysis of variance and Tukey’s post hoc test. The values are represented as mean ± SE.

**Figure 2 ijms-24-05815-f002:**
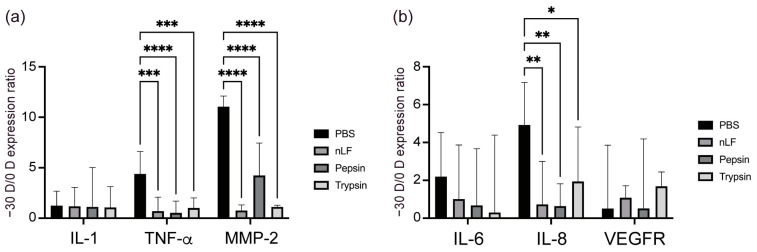
Changes in normalized gene expression levels in lens-induced myopia (LIM) mice administered lactoferrin (LF) and its derivatives. Calculations were performed using the ratio −30 D ∆∆CT/0 D ∆∆CT. (**a**) Interleukin (IL)-1, tumor necrosis factor (TNF)-α, and matrix metalloprotease (MMP)-2 mRNA expression level ratios in the sclera. (**b**) IL-6, IL-8, and vascular endothelial growth factor (VEGF) A mRNA expression level ratios in the choroid. n = 10 per group. * *p* < 0.05, ** *p* < 0.01, *** *p* < 0.001 and **** *p* < 0.0001. Comparisons between PBS, LF, pepsin + LF, trypsin + LF groups in each experiment were analyzed by analysis of variance and Tukey’s post hoc test. The values are represented as mean ± SE.

**Figure 3 ijms-24-05815-f003:**
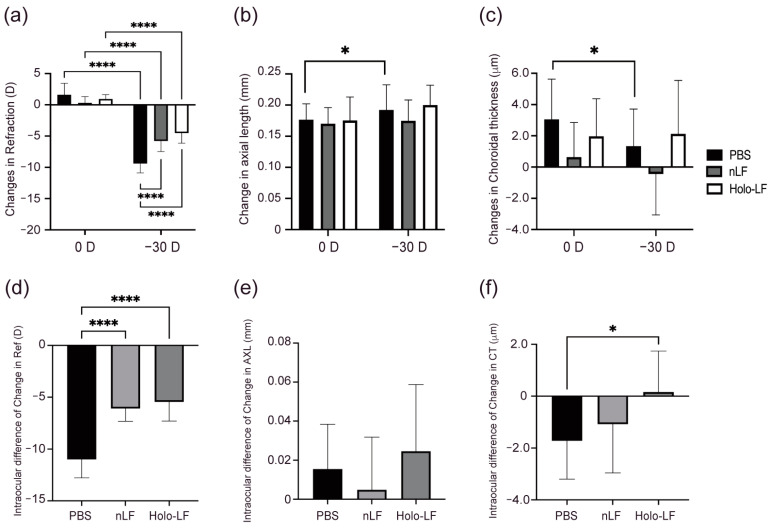
Changes in refraction, axial length (AL), and choroidal thickness (CT) in lens-induced myopia (LIM) mice administered native-lactoferrin (nLF) and Holo-LF. (**a**) Changes in refraction in 0 D and −30 D eyes over a 3-week period. (**b**) Changes in axial length (AL) in 0 D and −30 D eyes over a 3-week period. (**c**) Changes in choroidal thickness in 0 D and −30 D eyes over a 3-week period. (**d**) Delta between the refractive changes in 0 D and −30 D eyes over a 3-week period. (**e**) Delta between the AL changes in 0 D and −30 D eyes over a 3-week period. (**f**) Delta between the choroidal thickness changes in 0 D and −30 D eyes over a 3-week period. n = 10 per group. * *p* < 0.05, and **** *p* < 0.0001. The two-tailed Student’s *t*-test was used for left–right comparison analysis. Group comparisons were analyzed by analysis of variance and Tukey’s post hoc test. The values are represented as mean ± SE.

**Figure 4 ijms-24-05815-f004:**
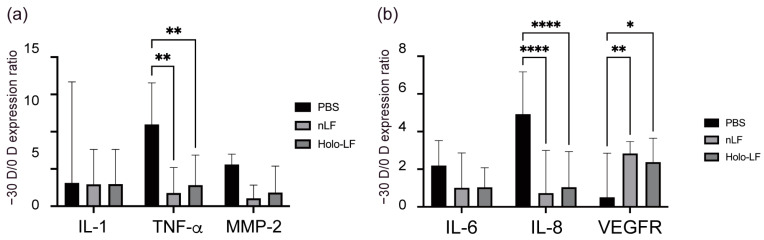
Changes in normalized gene expression levels in lens-induced myopia (LIM) mice administered lactoferrin (LF) and Holo-LF. Calculations were performed using the ratio −30 D ∆∆CT/0 D ∆∆CT. (**a**) Interleukin (IL)-1, tumor necrosis factor (TNF)-α, and matrix metalloprotease (MMP)-2 mRNA expression level ratios in the sclera. (**b**) IL-6, IL-8, and vascular endothelial growth factor (VEGF) A mRNA expression level ratios in the choroid. n = 10 per group. * *p* < 0.05, ** *p* < 0.01, and **** *p* < 0.0001. Group comparisons were analyzed by analysis of variance and Tukey’s post hoc test. The values are represented as mean ± SE.

## Data Availability

All datasets analyzed for this study can be found in the supplementary material files of this article.

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
