# Peer review of "Myopia Is Suppressed by Digested Lactoferrin or Holo-Lactoferrin Administration"

_ijms, 2023, doi:10.3390/ijms24065815_

Round 1
Reviewer 1 Report
All procedures performed on animals required surgical precision and were associated with a very high probability of eye contamination and infection. Therefore, I think that the authors of the publication should perform microbiological cultures of eye fragments collected during the autopsy. The more so that even small populations of bacteria could affect the level of pro-inflammatory cytokines, which were measured with very sensitive methods.
Author Response
Dear Reviewer 1,
Thank you very much for your careful review and thoughtful comments on our paper titled "Myopia is Suppressed by Digested Lactoferrin or Holo-lactoferrin Administration." We appreciate the time and effort you took to provide constructive feedback, and we believe your comments will help to improve the quality and impact of our work.
Point 1: All procedures performed on animals required surgical precision and were associated with a very high probability of eye contamination and infection. Therefore, I think that the authors of the publication should perform microbiological cultures of eye fragments collected during the autopsy. The more so that even small populations of bacteria could affect the level of pro-inflammatory cytokines, which were measured with very sensitive methods.
Response 1: Thank you for your constructive comments. We agree with you and think to incorporate this suggestion, however, the period we were given for the revise is 10 days and it is impossible to prepare an experimental system within that period (Since lactoferrin administration takes 4 weeks and myopia induction takes 3 weeks). However, since this proposal is a very important point of view for our future research, we plan to culture the bacteria, compare it with the results of this study, and report the results in a new resarch paper. We sincerely appreciate your very meaningful comments.
Reviewer 2 Report
Congratulates the authors on the idea. The research seems very pioneering. However, I have a few comments.
errors in preparing an article for a journal
References - The entire bibliography and in-text citations do not follow the journal's recommendations.
L12 – ‘’.’’ And L349 – ‘’ reagents..’’ - unnecessary periods at the end.
L34 – ‘’ Holo-Lactoferrin’’ - suggests writing in lowercase.
L448 – ‘’ Disclaimer/’’ - should be removed.
errors and omissions of substance
L35-45 - The authors describe myopia too little in the introduction. Why it is such an important problem for global society and what consequences it is associated with. In the introduction, the authors too shallowed the topic of myopia.
· Suggests adding probnostic data on myopia development.
‘’ Myopia and high myopia estimates from 2000 to 2050 suggest significant increases in prevalences globally, with implications for planning services, including managing and preventing myopia-related ocular complications and vision loss among almost 1 billion people with high myopia.’’
Holden B, Global Prevalence of Myopia and High Myopia and Temporal Trends from 2000 through 2050. Ophthalmology. 2016 May;123(5):1036-42.
· The authors describe that the eyeball is most often associated with myopia. This is, of course, true. However, there is a lack of clarification by how much , I would ask you to add.
‘’When grouping samples in 2 categories, a first peak appears around the AL of 24 mm for low myopia (–6 dpt < refractive error < 0 dpt) while the second peak appears roughly at the AL of 30 mm for high myopia (refractive error <–6 dpt)’’
Meng W, Axial length of myopia: a review of current research. Ophthalmologica. 2011;225(3):127-34. doi: 10.1159/000317072. Epub 2010 Oct 16. PMID: 20948239.
· Myopia is also an economic change, suggests adding information about the cost of treatment and economic problems associated with myopia.
‘’ Myopia is associated with substantial out-of-pocket expenditure, imposing considerable economic burden for patients. Myopia is a disorder with immense societal costs and public health impact.’’
Zheng Y, The economic cost of myopia in adults aged over 40 years in Singapore. Invest Ophthalmol Vis Sci. 2013 Nov 13;54(12):7532-7. doi: 10.1167/iovs.13-12795. PMID: 24159089.
And
Dutheil F, Myopia and Near Work: A Systematic Review and Meta-Analysis. Int J Environ Res Public Health. 2023 Jan 3;20(1):875. doi: 10.3390/ijerph20010875
· I also miss information in the introduction what refractive error is associated with myopia.
‘’ The current consensus threshold value for myopia is a spherical equivalent refractive error ≤ -0.50 diopters (D), but this carries significant risks of classification bias. The current consensus threshold value for high myopia is a spherical equivalent refractive error ≤ -6.00 D.’’
Flitcroft I, IMI - Defining and Classifying Myopia: A Proposed Set of Standards for Clinical and Epidemiologic Studies. Invest Ophthalmol Vis Sci. 2019 Feb 28;60(3):M20-M30. doi: 10.1167/iovs.18-25957.
· I suggest adding the possible consequences of myopia and the risk of diseases associated with it.
Haarman G, The Complications of Myopia: A Review and Meta-Analysis. Invest Ophthalmol Vis Sci. 2020 Apr 9;61(4):49. doi: 10.1167/iovs.61.4.49. PMID: 32347918; PMCID: PMC7401976.
And
Baird P,Myopia. Nat Rev Dis Primers. 2020 Dec 17;6(1):99. doi: 10.1038/s41572-020-00231-4. PMID: 33328468.
· I also suggest also giving away according to the latest research the possible impact of myopia on the musculoskeletal system.
Zieliński G, Masticatory Muscle Thickness and Activity Correlates to Eyeball Length, Intraocular Pressure, Retinal and Choroidal Thickness in Healthy Women versus Women with Myopia. J Pers Med. 2022 Apr 13;12(4):626. doi: 10.3390/jpm12040626.
And
Zieliński G, The Axial Length of the Eyeball and Bioelectrical Activity of Masticatory and Neck Muscles: A Preliminary Report. Pain Res Manag. 2022 Aug 16;2022:6115782. doi: 10.1155/2022/6115782
I think that developing the introduction with this information will be very beneficial and will increase the popularity of the article.
L62 - Add a clear objective and research hypotheses.
2. Results - The results are presented accurately, but I am missing the full ''p'' values. I propose to provide them for all results as supplementary material.
3. Discussion
The discussion is written in a correct manner. However, I suggest developing the discussion.
The authors focus on the effects of LFcin on ocular structures. It is one hundred correct and understandable. However, the organ of vision includes the extraocular muscles. They are responsible for movement and image stabilization. Current researchers emphasize two things the effect of LFcin on the muscular system (example papers below) and the connections between the muscular system and the organ of vision (papers I previously cited doi: 10.3390/jpm12040626 / 10.1155/2022/6115782). This is a hypothetical impact for now, but suggests developing a discussion on this topic.
Wang X, Lactoferrin Deficiency Impairs Proliferation of Satellite Cells via Downregulating the ERK1/2 Signaling Pathway. Int J Mol Sci. 2022 Jul 5;23(13):7478. doi: 10.3390/ijms23137478.
And
Kitakaze T, Lactoferrin promotes murine C2C12 myoblast proliferation and differentiation and myotube hypertrophy. Mol Med Rep. 2018 Apr;17(4):5912-5920. doi: 10.3892/mmr.2018.8603.
L227 - In my opinion, the conclusion is written correctly. Congratulations!
L247 – ‘’ Five mice’’ - ly test such a number of mice? Was it dictated by statistical calculations ?
L337 - I suggest adding an effect size and a confidence interval.
Sullivan GM, Feinn R. Using Effect Size-or Why the P Value Is Not Enough. J Grad Med Educ. 2012 Sep;4(3):279-82. doi: 10.4300/JGME-D-12-00156.1.
Suggests adding a list of abbreviations at the end of the paper.
Thank you for the opportunity to review.
With best regards
Author Response
Response to Reviewer 2 Comments:
Thank you for taking the time to review my paper. I appreciate your valuable feedback and suggestions, which have helped to improve the quality of my work. To facilitate your review of our revisions, the following is a point-by-point response to the questions and comments.
Comment 1: References - The entire bibliography and in-text citations do not follow the journal's recommendations.
Response 1: We updated the format of citations with following the IJMS’ recommendation.
Comment 2: L12 – ‘’.’’ And L349 – ‘’ reagents..’’ - unnecessary periods at the end.
Response 2: We removed the unnecessary periods at the end.
Comment 3: L34 – ‘’ Holo-Lactoferrin’’ - suggests writing in lowercase.
Response 3: We changed “Holo-Lactoferrin” to “holo-lactoferrin”.
Comment 4: L448 – ‘’ Disclaimer/’’ - should be removed.
Response 4: We removed “Disclaimer/”.
Comment 5: L35-45 - The authors describe myopia too little in the introduction. Why it is such an important problem for global society and what consequences it is associated with. In the introduction, the authors too shallowed the topic of myopia.
Suggests adding prognostic data on myopia development.
‘’ Myopia and high myopia estimates from 2000 to 2050 suggest significant increases in prevalences globally, with implications for planning services, including managing and preventing myopia-related ocular complications and vision loss among almost 1 billion people with high myopia.’’
Holden B, Global Prevalence of Myopia and High Myopia and Temporal Trends from 2000 through 2050. Ophthalmology. 2016 May;123(5):1036-42.
Response 5: We agree with you and have incorporated this suggestion throughout the introduction of our paper.
We added a sentence and citation to introduce the trend of myopia development.
“As an estimate of the global prevalence of myopia, the prevalence of myopia and high myopia is expected to rise significantly on a global scale, affecting almost 5 billion and 1 billion people, respectively, by the year 2050 [2].”
Comment 6: The authors describe that the eyeball is most often associated with myopia. This is, of course, true. However, there is a lack of clarification by how much , I would ask you to add.
‘’When grouping samples in 2 categories, a first peak appears around the AL of 24 mm for low myopia (–6 dpt < refractive error < 0 dpt) while the second peak appears roughly at the AL of 30 mm for high myopia (refractive error <–6 dpt)’’
Meng W, Axial length of myopia: a review of current research. Ophthalmologica. 2011;225(3):127-34. doi: 10.1159/000317072. Epub 2010 Oct 16. PMID: 20948239.
Response 6: We agree that we need to clarify how much is the axial length associated with myopia, however, due to the description “peak appears roughly at the AL of 30mm” of the reference paper is an aged data, we believe that we need to introduce this association with recent studies.
Here, we added sentences and citations to introduce the relationship between myopia and axial length.
“Low myopia, with a refractive error between –6 D and -0.5 D, has an axial length between 24 mm and 26 mm, while high myopia, with a refractive error lower than –6 D, has an axial length longer than 26 mm [4-6]. Reports have consistently shown a negative relationship between axial length and myopia, indicating that longer axial lengths are associated with more severe myopia [7-9].”
Comment 7: Myopia is also an economic change, suggests adding information about the cost of treatment and economic problems associated with myopia.
‘’ Myopia is associated with substantial out-of-pocket expenditure, imposing considerable economic burden for patients. Myopia is a disorder with immense societal costs and public health impact.’’
Zheng Y, The economic cost of myopia in adults aged over 40 years in Singapore. Invest Ophthalmol Vis Sci. 2013 Nov 13;54(12):7532-7. doi: 10.1167/iovs.13-12795. PMID: 24159089.
And
Dutheil F, Myopia and Near Work: A Systematic Review and Meta-Analysis. Int J Environ Res Public Health. 2023 Jan 3;20(1):875. doi: 10.3390/ijerph20010875
Response 7: This is an interesting perspective, and we totally agree with that.
We added a sentence and citation to introduce the information about the cost of treatment and economic problems associated with myopia.
“In addition, myopia in Singapore, for example, costs SGD$959 (USD$755) million annually, resulting in significant out-of-pocket expenses that burden patients and their families; thus, refractive correction has a notable public health and economic impact [14].”
Comment 8: I also miss information in the introduction what refractive error is associated with myopia.
‘’ The current consensus threshold value for myopia is a spherical equivalent refractive error ≤ -0.50 diopters (D), but this carries significant risks of classification bias. The current consensus threshold value for high myopia is a spherical equivalent refractive error ≤ -6.00 D.’’
Flitcroft I, IMI - Defining and Classifying Myopia: A Proposed Set of Standards for Clinical and Epidemiologic Studies. Invest Ophthalmol Vis Sci. 2019 Feb 28;60(3):M20-M30. doi: 10.1167/iovs.18-25957.
Response 8: You have raised an important point.
We added a sentence and citation to introduce the relationship between myopia and spherical equivalent refractive error.
“A spherical equivalent refractive error of -0.50 diopters (D) or less is the current consensus threshold value for myopia, while a spherical equivalent refractive error of -6.00 D or less become the threshold value for high myopia [4], which can increase the risk vision impairment.”
Comment 9: I suggest adding the possible consequences of myopia and the risk of diseases associated with it.
Haarman G, The Complications of Myopia: A Review and Meta-Analysis. Invest Ophthalmol Vis Sci. 2020 Apr 9;61(4):49. doi: 10.1167/iovs.61.4.49. PMID: 32347918; PMCID: PMC7401976.
And
Baird P,Myopia. Nat Rev Dis Primers. 2020 Dec 17;6(1):99. doi: 10.1038/s41572-020-00231-4. PMID: 33328468.
Response 9: We have reflected this comment to emphasize the complication of myopia.
“Owing to morphological changes, physical stimuli such as stress are applied to the posterior segment of the eye, resulting in visual impairment, such as myopic macular degeneration (MMD), retinal detachment (RD), cataract, and open angle glaucoma (OAG),and macular optic neuropathy [3,10,11].”
Comment 10: I also suggest also giving away according to the latest research the possible impact of myopia on the musculoskeletal system.
Zieliński G, Masticatory Muscle Thickness and Activity Correlates to Eyeball Length, Intraocular Pressure, Retinal and Choroidal Thickness in Healthy Women versus Women with Myopia. J Pers Med. 2022 Apr 13;12(4):626. doi: 10.3390/jpm12040626.
And
Zieliński G, The Axial Length of the Eyeball and Bioelectrical Activity of Masticatory and Neck Muscles: A Preliminary Report. Pain Res Manag. 2022 Aug 16;2022:6115782. doi: 10.1155/2022/6115782
I think that developing the introduction with this information will be very beneficial and will increase the popularity of the article.
Response 10: Thank you for your suggestion and we think that this an interesting perspective.
We added a sentence and citation to introduce the relationships between myopia and musculoskeletal system.
“The visual system comprises the extraocular muscles, which play a vital role in both moving the eyes and stabilizing the resulting images. Myopia is also proved that associated with masticatory muscle bioelectrical activity and muscle thickness [12,13].”
Comment 11: L62 - Add a clear objective and research hypotheses.
Response 11: We have incorporated your comments by adding a sentence to introduce our objective and research hypotheses in this study.
“To investigate the differences in the suppressive effects of native-LF and LF derivatives on myopia progression, we designed the present study using the described LF products. We predicted that the LF derivatives could suppress the myopia progression more efficiently.”
Comment 12: 2. Results - The results are presented accurately, but I am missing the full ''p'' values. I propose to provide them for all results as supplementary material.
Response 12: We will provide the full “p” values of all results in table as supplementary material in an Excel file.
Comment 13: 3. Discussion
The discussion is written in a correct manner. However, I suggest developing the discussion.
The authors focus on the effects of LFcin on ocular structures. It is one hundred correct and understandable. However, the organ of vision includes the extraocular muscles. They are responsible for movement and image stabilization. Current researchers emphasize two things the effect of LFcin on the muscular system (example papers below) and the connections between the muscular system and the organ of vision (papers I previously cited doi: 10.3390/jpm12040626 / 10.1155/2022/6115782). This is a hypothetical impact for now, but suggests developing a discussion on this topic.
Wang X, Lactoferrin Deficiency Impairs Proliferation of Satellite Cells via Downregulating the ERK1/2 Signaling Pathway. Int J Mol Sci. 2022 Jul 5;23(13):7478. doi: 10.3390/ijms23137478.
And
Kitakaze T, Lactoferrin promotes murine C2C12 myoblast proliferation and differentiation and myotube hypertrophy. Mol Med Rep. 2018 Apr;17(4):5912-5920. doi: 10.3892/mmr.2018.8603.
Response 13: Thank you very much for your suggestion.
Here we added a paragraph to discuss the relationship between myopia, LF and muscular system.
“The visual system, including the extraocular muscles, plays a critical role in both moving the eyes and stabilizing resulting images. Recent Studies suggest that myopia may be associated with masticatory muscle bioelectrical activity and muscle thickness [12,13]. On the other hand, LF may act as a functional factor that maintains the proliferation of stem cells (SCs) and promotes myoblast proliferation by activating the extracellular signal-regulated kinase (ERK)1/2 signaling pathway, at least partially through LRP1, and inducing the differentiation of myoblasts into myotubes. In addition, LF promotes myotube hypertrophy. LF depletion impaired the proliferation of SCs by downregulating ERK1/2 signaling. Moreover, intraperitoneal injection of exogenous recombinant lactoferrin (R-LF) was found to effectively help repair and renew skeletal muscle after injury. Thus, LF may suppress myopia progression by repairing the function of the muscular system [33,34].”
Comment 14: L227 - In my opinion, the conclusion is written correctly. Congratulations!
Response 14: Thank you for your appreciation.
Comment 15: L247 – ‘’ Five mice’’ - ly test such a number of mice? Was it dictated by statistical calculations?
Response 15: The sample size of each group could be calculated to be 4. Thus, we used 5 mice per group to perform the experiment and this number of mice is totally enough. We have also redrafted the animal section (p. #7, lines #275-#278) to establish a clearer focus.
“In each group, five mice were used in the experiment, and the whole experiment was repeated once to ensure reproducibility. This number is calculated based on previous study [26] with a 2.67 effect size and a 95% confidence interval by G Power (Version 3.1.9.7, University Dusseldorf, Germany).”
Comment 16: L337 - I suggest adding an effect size and a confidence interval.
Sullivan GM, Feinn R. Using Effect Size-or Why the P Value Is Not Enough. J Grad Med Educ. 2012 Sep;4(3):279-82. doi: 10.4300/JGME-D-12-00156.1.
Response 16: Thank you for providing this important insight. We calculated the effect size with a 95% confidence interval, and the result was 2.67. Effect size and sample size were both calculated by G Power (Version 3.1.9.7, University Dusseldorf, Germany).
Comment 17: Suggests adding a list of abbreviations at the end of the paper.
Response 17: Thank you for your suggestion. We added a list of abbreviation.
In addition to these specific revisions, we have also carefully reviewed and edited the manuscript for clarity, accuracy, and readability.
We believe that these revisions have significantly strengthened the manuscript, and we are confident that it now meets the high standards of your journal. We are grateful for the opportunity to revise and resubmit, and we hope that the revised manuscript will be acceptable for publication in your journal.
Thank you again for your valuable feedback and your consideration of our work. We look forward to hearing from you.
Round 2
Reviewer 2 Report
The authors correctly responded to my suggestions. I recommend the papers for publication. Congratulations.